# Evidence of the Relationship between Social Vulnerability and the Spread of COVID-19 in Urban Spaces

**DOI:** 10.3390/ijerph19095336

**Published:** 2022-04-27

**Authors:** Federico Benjamín Galacho-Jiménez, David Carruana-Herrera, Julián Molina, José Damián Ruiz-Sinoga

**Affiliations:** 1Geographical Analysis Group, Department of Geography, University of Malaga, 29071 Málaga, Spain; 2Departament of Geography, University of Malaga, 29071 Málaga, Spain; carruana@uma.es; 3Department of Applied Economics, University of Malaga, 29071 Málaga, Spain; julian.molina@uma.es; 4Physical Geography and Territory Group, Department of Geography, University of Malaga, 29071 Málaga, Spain; sinoga@uma.es

**Keywords:** COVID-19, social vulnerability, social areas, socio-spatial structure, spatial analysis, GIS

## Abstract

Modeling the social-spatial structure of urban spaces can facilitate the development of guidelines aimed at curbing the spread of the COVID-19 pandemic while also acting as an instrument that helps decision-making concerning mitigation policies. The modeling process starts with categorization of urban spaces based on the concept of social vulnerability. A model is created based on this concept and the theory of analysis of social areas. Statistical techniques of factor analysis and geostatistics are applied. This generates a map of social differentiation that, when related to data on the evolution of the contagion, generates a multidimensional model of social vulnerability. The application of this model towards people (social structure) and the environment where they live (spatial structure) is specified. Our model assumes the uniqueness of cities, and it is intended to be a broadly applicable model that can be extrapolated to other urban areas if pertinent revisions are made. Our work demonstrates that aspects of the social and urban structures may be validly used to analyze and explain the spatial spread of COVID-19.

## 1. Introduction

Urban spaces have suffered from the spread of the COVID-19 pandemic in very different ways, with virulence often depending on the social fabric, diversity, and complexity of an urban space [1,2,3]. Here, we start with the belief that there is a direct relationship between social and spatial structures that can be synthesized through the concept of social vulnerability, and that this impacts the intensity and forms of contagion distribution in urban spaces. By relating these concepts to one another, we hope to describe and explain the conditions underlying the spread of the pandemic.

Toward this end, we carry out a multidimensional analysis and propose a spatial model that allows the representation of the vulnerability condition together with the COVID contagion situation [4,5]. We consider that representing together these two features could be useful to design prevention policies, although we assume the relation between COVID contagion and vulnerability is not universal [6]. We intend for our model to be usable in other geographical environments, but previous studies in Spanish urban spaces [7,8,9,10] suggest that it would need to be redefined to some degree for use elsewhere. The model also cannot be applied with similar suitability to cities of all sizes. Cities have different characteristics and functions due to their sizes, cultural differences, expansion patterns, and/or other circumstances. The social structure of each urban area reflects a synthesis of these particularities [11,12]. It can thus seem very complex to extract a general theory, generate a broadly applicable method, or simply establish an absolute explanation of the facts that is based on an analysis of the issue at hand and can be extrapolated to other areas. In short, the specificity of each urban area imposes an unavoidable singularity [9,13,14,15].

This work seeks to meet this challenge. Analyzing the pandemic by characterizing the spatial differentiation of social groups is an open field of research. The application of the obtained and validated results is of great interest for the design of current and future management strategies. From the methodological point of view, the social characterization of the urban space is first addressed by selecting the characteristics of the population, as they form part of the analytic method. These characteristics are collected into categories that define their generic characteristics as a whole and, especially, the particularities or specifics of their organization. These categories are then used to define the mosaic of social worlds in which the urban space is articulated [16]. Next, the characterization of COVID-19 infections is carried out. Finally, the relationship between the social and spatial typology and the spatial distribution of contagion cases is studied and used to create an index of social vulnerability for the social areas of the urban space.

The first references to the concept of the social dimensions of vulnerability appear in studies on health sciences, the prevention of infectious diseases such as HIV/AIDS [17,18,19], and other aspects of population health [20,21]. The use of this concept has since spread to other fields, including those that work with both spatial concepts and natural risks for a population [22,23] and social sciences within the fields of social trajectories, interactions, and social context (e.g., studies on crime and racial harassment) [24]. Recently, along the same lines as the present work, the social dimensions of vulnerability have been assessed in the field of health geography and spatial analysis [25]. These works have added new dimensions to the concept. However, certain difficulties have arisen when researchers have sought to apply the social dimensions of vulnerability to real situations in which individuals or groups of individuals are more highly exposed to infectious diseases. Therefore, herein, we seek to contextualize the concept of vulnerability and reinforce its heuristic capacity and political and practical relevance [26,27].

As COVID-19 causes severe acute respiratory syndrome-coronavirus 2 (SARS-CoV-2) [28,29,30,31,32,33], researchers have investigated the effects of environmental conditions, including the effects of air pollution (for example: PM2.5 particles) and meteorological variables (e.g., prevailing temperatures) on the transmission of the disease. Other studies have addressed the transmission of SARS-CoV-2 pathogens in urban contexts associated with seasonal climate variability and drivers of air pollution [34,35], as well as in different geographical/morphological contexts, such as rural spaces versus small cities [36].

We analyze the spread of COVID according to the population’s conditions. If we focus exclusively on applying our model to people under the concept of social structure, our analysis will be incomplete: We must also observe the environment in which social groups live. When we consider the two aspects together, we get what we call the “spatial structure”, which is a mosaic of the social worlds in which an urban space is articulated. We take this approach because it is understood that people who are most exposed to risk will be further exposed to contagion by the indirect mechanisms, and thus they have to protect themselves. This will be synthesized by considering the spatial context that encompasses them.

We start from the hypothesis of an unequal effect of the pandemic for the different areas, by visualizing the unequal incidence of COVID-19 disease in different urban spaces of the city of Malaga, Spain. In this geographic space, we apply a multidimensional analytic methodology based on defining four categories: three for social characterization, demographic, socioeconomic, and social care; and one for spatial characterization, territorial. Various recent reports have analyzed the influence of socioeconomic conditions on the living conditions and health of populations [37,38,39] and the role of socioeconomic factors in the spread of the pandemic and its social consequences [40,41,42,43,44,45]. The local and intercity transmissions of COVID-19 have also been analyzed in studies examining the role of socioeconomic factors and public health measures imposed by health authorities [46,47].

Our model seeks to obtain and measure the main society’s features to be related with the COVID spread, i.e., the aspects of the studied regions’ social structure, and the spatial structures that identify it as a modern society and provide the configuration of its social scale. The resulting socio-spatial configuration will be related to the number of contagion cases to observe their distribution in the urban space, with the intention of determining the possible propagation patterns. The model is developed based on the scheme shown in Figure 1.

## 2. Materials and Methods

### 2.1. Study Area

As an experimental area for the application of the designed model, we chose the urban area of the city of Malaga, Spain. The urban area of this city covers an area of 94.46 km^2^ and had a population of 574,251 inhabitants in 2020. Regarding COVID-19, a total of 25,479 cases occurred up to March 2020, 2447 (9.60%) of which required hospitalization; see Figure 2.

The units known as “districts” are an administrative division of the city that collect various historical–social transformations and the changes that have generated socio-cultural tensions among inhabitants and/or modified the meaning of the lived space (physical, ways of inhabiting it, and its consequent social relations), thereby crystallizing the different ways in which inhabitants relate to one another and with their space. It is common to find fragmentation of the social fabric and different patterns of neighborhood behavior. We can observe how the inhabitants in the different spaces appropriate the space, identify themselves, become attached, orient in common practices, and fight among themselves to imprint their vision and way of inhabiting said space. In these units, it is possible to recognize and understand the objective structure of social groups and identify their characteristics. We can also observe the representations and social practices that they build with their appropriation of space. This is very important in the lived context of the COVID-19 pandemic because it allows us to observe the modes of participation and how the population unfolds in its immediate environment. This further enables us to identify the characteristics that configure the behaviors, including the cultural orientations that dominate the objective structure of these spaces, their cognitive aspects, and the situational context. In Table 1, we show the 11 districts/urban social areas in which the variables are analyzed and then related to the level of COVID-19 infection.

The districts synthesize information on the urban space through a selection of socially significant statistical, demographic, and economic data. This allows territorial disaggregation, which enables us to observe the complexity of the urban space. Having social information disaggregated by these geographical areas is essential for our ability to understand the city, so the well-known names for these districts have been respected instead of using a list of generic names. Knowing the socio-spatial structure of this space is a necessary starting point.

### 2.2. Data Collection

For the social and territorial features, we are going to use the data on a complete and precise vulnerability analysis of the city of Málaga, conducted by the CIEDES Foundation, in collaboration with the Malaga City Council and the University of Malaga, where they examined the vulnerability of the different neighbourhoods of Malaga in multiple dimensions [48]. This study generated a global vulnerability index that consisted of the average of 4 vulnerability sub-indices, related with the four main social and environmental characteristics of a population: family status, social status, social care, and spatial framework. To build these four sub-indices, 19 variables were selected from several official sources and from a short-term survey on living conditions, as shown in Table 2. These variables included from dependency rate, aging index or household income to unemployment rate, social integration needs, green zones, accessibility, or home size. We will refer to the selected variables in detail in the results section.

On the other hand, the contagions data comes from the Andalusian Health Service (SAS) of the Andalusia’s regional govern. In addition to infections, this database provides us with the date of infection, complete address, age, gender, type of test performed (PCR or antigens), if the patient was admitted to the hospital, and finally, if they died. So, for each district, we are going to compare the information of the COVID contagions with the global vulnerability index and with each of the four vulnerability sub-indices, whose values are summarized in (Table 3). All the indices are ranged in [0,1], both the global and the sub-indices, and for all of them the closer to 0 means less vulnerability, and the closer to 1 means more vulnerability.

### 2.3. Analysis of the Data on COVID-19 Infections in the Space Analyzed

The spread of COVID-19 has followed different patterns based on the context of the spread and the health actions developed in each country. Thus, it is crucial that we identify what stage of spread the epidemic has reached in a given country and indicators of relative variability of confirmed cases [48,49,50]. Our statistical analysis reveals that the characteristics of the social areas have the expected correlations with the number of cases in relation to the variables selected in each category of the model. In the city of Malaga, differences are observed according to social areas (see the distribution by area in Table 4). Similar situations have been found in other cities of Spain and in other countries [51,52].

Figure 3 shows the variability of the number of infections depending on the age group and according to the spaces of the analyzed urban context.

The idea that is proposed for reflection with Table 4 and Figure 3 is the extent to which the differences in the spatial distribution of infections by districts is determined by the age structure of their population and the extent to which the differences in the number of cases are supported by the processes of social segregation. It is observed that the intensity of infections is conditioned by the intensity of social segregation in the districts. The expansion of the number of cases has been conditioned to the spatial differences both more appreciable and accelerated and wide has been the intensity of the pandemic. What has been observed suggests that one of the factors that has differentiated this process has been the age structure, although it is not the only one. The peripheral districts, such as Campanillas, Puerto de la Torre, Teatinos-Universidad, or Churriana, have suffered the pandemic with less intensity due to the characteristics of their populations. However, in the interior space of the city, i.e., Carretera de Cádiz, Bailén-Miraflores, Cruz de Humilladero, and Centro, the pandemic has been felt with greater intensity. The latter are areas of the traditional city where very defined social spaces have been generated. Its interior homogeneity is more complex and socially intermingled, and its urban consolidation began in the 1940s, in a dynamic of slow processes of expansion, compaction, and renovation. The natural processes of physical deterioration of the urban space have led, due to its progressive devaluation, to the prevalence of less affluent social classes and older population groups. Subsequently, the urban space is structured in areas of youth and different levels of social inequality in a very regular sequence from the center to the periphery as has been the temporary process by which the pandemic has been extended. Therefore, this evidence is one of the explanatory factors of the spatial distribution of the number of infections in the different waves that have occurred. That is why it is one of the relevant aspects to consider. We analyze it below.

On the other hand, the spatial distribution of infections is analyzed by applying a kernel-type geostatistical analysis model (see Figure 4). This density analysis takes the known locations of infections and the number of cases measured at each location and uses them to create the spatial relationship of the measured quantities. The density surfaces show where the largest number of points representing the total number of infected people are concentrated [53,54].

This leads us to speculate that this heterogeneity is a main cause in the existence of differential spatial behaviors during different waves of the pandemic, as expected. The graphic representation enables us to highlight this important fact and visualize how there have been more cases of contagion in some areas than in others. We have carried out a process of identifying the existence of geographical patterns of the distribution and special association of the number of cases. The application of this method is aimed at the evaluation of the mode of implantation, implying that the spatial heterogeneity derived from said implantation can condition the territorial distribution of the number of cases, in which agglomerates are now formed through a densification dynamic that arises in a process not defined, but random. It can be understood that there are more cases of contagion in some areas than in others, which is logical and could be expected, but this study is precisely about demonstrating that this fact happens and how it is configured spatially. We must analyze the distribution of contagion cases with reference to the spatial location where they occurred, as this enables us to make comparisons related to the category-level characteristics of the social areas.

## 3. Results

In the analysis of the social structure, we refer to the first category used herein to identify the scale of society is family status (Figure 5). Through this category, we can observe the differentiation of functions from the demographic point of view. In the structure of the social system, the incidence of the family status category is specified in the configuration of life models that generate characteristic family situations.

In this sense, within the set of variables considered determinants for the model according to the object of this work we highlight, related to family status, the variables: people over 75 years of age who live alone, the dependency rate and the ageing index help explain the behavior of the category. These indicators collect data on the active population and the dependent population, and we can see the aging process. In Spain, people who live alone tend to have less favorable living conditions, especially as their age advances; the effects of less favorable living conditions combine with other contextual aspects, such as health and household structure [55]. Among people with health problems or sensory deficiencies, the appearance of new symptoms or the aggravation of existing ones may go unnoticed. Many elderly people in these circumstances find it difficult to comply with prescribed therapeutic regimens. Given that many elderly people have physical limitations and eating is a social activity, some elderly people who live alone do not prepare complete and balanced meals. Malnutrition, which is a frequent problem in this population [56], increases the vulnerability to infectious diseases.

The second basic category is social status (Figure 6). This category expresses the change in the order and intensity of relationships in urban spaces, which manifests in a decreased importance of productive activities and a considerable increase in activities related to administration, management, and services. In the social structure, this implies important changes in the scale and organization of jobs and, consequently, differential access to labor income. We hypothesize that the behavior of the pandemic may be configured at least partly by social inequality and/or the way in which urban dynamics model the differentiation of economic characteristics over space. We consider that the consequences of socioeconomic inequalities are transmitted to the health of the population. Wilkinson and Pickett [57,58] proposed a series of consequences associated with inequality, including a lack of material goods, more limited educational opportunities, a lower probability of ascending the social scale, and issues such as decreases in physical and mental health. Several studies have addressed these theories [59,60,61].

In the category of social status, the following variables have been considered: household income, illiterate or uneducated population, job seekers and work intensity. The first two are considered decisive and the second two complementary. Of these, household income refers to quality of life and can be measured in terms of disposable income. With this information, it is possible to know the standard and conditions of living, social cohesion, social protection, and the poverty level of households. Differences in the level of income can lead to the incidence of the pandemic decreasing when citizens have the possibility of living in less dense spaces and further away from congested areas. It is noted that social costs have appeared with origin in the different forms of the city: households of social groups with lower incomes are forced to concentrate in more densified spaces, where housing is smaller and cheaper, especially in relation to the number of people such households have to accommodate. Regarding the second variable, it is known that illiteracy and, mainly, functional illiteracy (people who have basic knowledge of reading, writing, and calculation, but who are not able to use this knowledge efficiently in situations of daily life), in addition to limiting the full development of people and their participation in society, has repercussions throughout its life cycle, affecting the family environment, restricting access to the benefits of development and hindering the enjoyment of other rights, having significant personal, intergenerational, social, and economic consequences.

As noted above, measuring the quality of life based on income provides information on the levels and conditions of life, social cohesion, social protection, and poverty for households. An increase in income level can decrease the incidence of the pandemic when citizens have the opportunity to live in less dense spaces and farther away from congested areas. Our analysis confirms that there are different social costs in the different urban areas: The households of lower-income social groups are forced to concentrate in denser spaces where the dwellings are cheaper and small in relation to the number of people who need to be accommodated.

The social structure of the model is completed with the category of social care (Figure 7). This category shows the intensity of access to social services and is based on variables, such as the number of people received in attention centers, the need for social integration, and the resources applied for subsistence needs. The year 2020 was marked by the health crisis caused by COVID-19, which in the field of dependency management required exceptional measures to contain the contagion. We include the care perspective to account for the way in which social care can be used to minimize the effects of the pandemic. Our analysis reveals that this category contributes to explaining the behavior of the model but is not decisive.

Finally, we include the territorial framework, which is a category for the spatial structure (Figure 8). An environment can be fundamentally characterized based on the parameters that define the conditions imposed by the spaces in which different social groups live. After the COVID-19 pandemic appeared and spread in Spain, clear differences were observed from the spatial point of view in terms of the urban spaces, both in each Spanish city (as studied herein) and between/among Spanish cities. These differences affect both infections and the possibilities of complying with the measures imposed to alleviate the pandemic. Therefore, this category is leveraged to determine if the conditions of the urban environment are statistically associated with the spread of COVID-19, while seeking to avoid possible relationships that are not clear and/or that hide the real significance of the analyzed variables. For this, different multiple regression models are proposed. In addition to the variable of interest, a wide set of other variables that may be relevant to disease propagation are introduced in each case. These variables must also allow us to delve into the differences in the scales of the social area and the city. Although we circumscribe our analysis to urban spaces, such spaces may have natural conditioning factors (e.g., variables related to microclimatic conditions) that favor the different valuation of social areas. In some cases, this may reflect a direct effect of the living conditions, while in other cases it may reveal how the natural conditioning factors affect the spread of the pandemic. We found relatively few published studies that have introduced the variables that we use herein, either individually or separately [62], and therefore submit that these variables have not been sufficiently considered in the analyses carried out to date.

The data of the category of the territorial framework may provide insight into the spread of COVID-19 according to the urban structure and the forms of mobility of the population. In urban areas with high building density and a dense urban fabric, the average size of the dwelling in relation to the number of family members will favor a situation of overcrowding and precariousness (Figure 8). This will increase the possibility of disease propagation and make it impossible to comply with confinement measures [63]. In this context, we note that very few studies have proposed adequate strategies for the design of urban spaces that facilitate the fight against pandemics [64].

According to the selected categories (family status, social status, social care, and territorial framework), we calculate a social impacts index. We seek to highlight the substantial relationships between the social and family structures of the space and the evolution of the COVID-19 pandemic. Above, we describe the processes that allowed us to conceptualize the social and spatial structures based on the categories analyzed and consider the contagion expansion processes based on socio-spatial differentiation. Now, we bring them together to look at the evolution of the pandemic in the urban spaces of Malaga, with its differentiated social spaces. All of this is specified in the analysis of social vulnerability according to the categories of our model. From this perspective, differentiation is not strictly economic (although we undoubtedly reveal a very close relationship), but also relates to the social valuation of space as an indicator of the different outcomes for social groups under the pandemic.

The final impact map, which represent the sum of the four categories (Figure 9), provide additional insights regarding the social areas identified as having very low social status. The highest values of this index indicate situations characterized by a high prevalence in areas of historical urban development, which have experienced greater impacts of the pandemic. Conversely, areas with the highest social status exhibit the lowest impact values. Between these extremes lie spaces with intermediate vulnerability and a range of impact values.

As can be seen in the Figure 9, many of the social areas located in historic urban developments (Ciudad Jardín, Palma-Palmilla; Bailén-Miraflores y Cruz de Humilladero) are among the most highly affected by the pandemic. These areas, without being on the extreme vulnerability, are on the highest vulnerability segments.

In these areas the values of the vulnerability index are in the highest ranges. The acquired level of this phenomenon can be related to these are areas of urban space in which pockets of urban deterioration are the triggers of its most recent social differentiation. Thus, in these districts coexist social classes of medium social status along with marginal ghettos. Therefore, it is important to consider the level of social inequality in these areas as a possible trigger for the expansion of the number of COVID-19 cases. They are areas in which the behavior of the pandemic can be related to a homogeneous character of urban society, where the most overwhelming weight of the urban middle classes extraordinarily nuances the intensity with which the processes of expansion of the pandemic are reflected in social areas differentiated by their social status.

In terms of the professional structure, the populations of these areas identify work intensities across all (low, medium, and high) intervals. The job-seeker variable fluctuates in these areas because it depends on the required level of education, which ranges from low (manual activities such as construction and basic services) to high (e.g., management and security). All this configures a situation that we can relate to a greater number of infections that we can relate to the social deterioration of these spaces.

In the urban periphery and areas of recent urban development (Carretera de Cádiz, Churriana, Campanillas, and Puerto de la Torre). In contrast, the vulnerability level is medium–low, the life cycle is relatively young, and there are typically fewer infections.

In these districts, social characterization is determined by the patterns of residential mobility that respond to the search for the satisfaction of the needs, or aspirations, of the population groups in terms of residence. This fact can be related to the life journey, to changes in family and social status, and according to economic possibilities. In this situation, without ignoring the factors linked to social status that also determine mobility, it is the changes in family status that become most clearly perceptible. It could be said that, as they are new urban areas, the most mobile population is based in them, that of the new families of medium life cycle (young adults), in phases of overcoming or consolidating their economic status. Consequently, these urban spaces attract moderately young families, in relation to a mobility more linked to the overcoming of status than to the initial settlement when forming a new family, so they are structured in areas of youth.

In social areas with higher social status (Este), meanwhile, the life cycle may be relatively old, but the weight of this parameter is reduced because of the living conditions and dwelling habitability, leading to the observation of relatively few cases of contagion compared to the historical urban development sectors. These are typically minority areas and may be populated by aging groups living in areas of historical urban development or young populations in areas of the urban periphery. At the other extreme is the population of high social status areas, which differ profoundly from the other social areas. Of the 11 social areas that we identify in the urban space of Malaga, only one has a unique set of characteristics: The so-called area of East Malaga has an insignificant population of illiterate or low-education status individuals, high social status, the highest work intensity, and one of the highest rent levels, reflecting larger homes with more free spaces. These features define a social level that is very advanced with respect to the average of the studied social areas. Spatially and with reference to the pandemic, East Malaga has a lower number of infections than the average. It is a peripheral area of high social status. It stands out for its spatial features showing a great diversity of building typologies, in an environment of high environmental quality and characterized by a low or moderate urban density. That is why, in this area, the well-off urban classes have settled in replacement of the attraction previously exerted on them by the central areas of the urban space. The social groups that have been attracted to this area manifest a collective appetite to enjoy a quality of life that is defined by the lower constructive density and the spaciousness of the house. Moreover, it cannot be ignored that this is achieved by its purchasing power.

By comparing the distribution of COVID-19 cases with the categories used to characterize the social areas, we reveal a complex and outsourced society in which the analyzed variables point to clearly differentiated social structures and an evident relationship between these structures and the volume of infections. First, there is a tendency towards aging in societies of the type of Málaga. An increase in the acute aging of a population is only compensated by the arrival of new individuals through immigration. The age of a society is a crucial issue in the development of the COVID-19 pandemic. Areas with fewer infections often have younger demographics and thus lower aging rates, which can offset the weight of people over 75 years who live alone. By considering young versus aged spaces, we can differentiate urban spaces in areas with an individualized social profile. In the case of Malaga, the so-called urban peripheries and those of recent urban development present a predominance of younger populations and tend to reflect lower levels of contagion. The weight of the child or youth population, which has been less affected by the pandemic, means that these areas appear to be less-affected spaces. These areas are also often characterized as being a place of residence or residential mobility for people in the middle life cycle, who often represent young families with minor children and a higher income level. These incoming families bring youth to the population structure, but they have care needs that must be addressed alongside those of the natives. This is captured in the care framework of our model.

The different types of impacts affected populations are analyzed using a Pen’s Parade curve [65,66], as presented in Figure 10. We see the value of the vulnerability index (vertical axis) for each cumulative population percentile (horizontal axis) and can follow the distribution of vulnerabilities across the population. More horizontal curves indicate a homogeneous distribution of vulnerability, while steeply sloped sections indicate notable changes in the vulnerability level.

Figure 10 reveals that social care impact is not evenly distributed. High percentages of the population have a similar level, while a small percentage has a very high impact, causing the curve to become vertical in its last section. At the other extreme, we see that the influence of territorial impact is more evenly distributed and homogeneous across the population. Social impact, which presents a steep final slope, responds to the disparity of social status in the analyzed urban space.

The differences found in the COVID-19 impacts maps for each category also show this segregation, which is essentially explained by the ability of groups with the highest economic capacity to select a space to live in, and the relative lack of living-space choices for those of lower social status. This concept is consistent with the findings of previous studies [67,68,69].

Note that there is a difference in the use of terms. The rating of those with the highest economic capacity is configured based on parameters that are more stable (or less vulnerable), while those of lower social status experience convergence of variables beyond the economic one, narrowing their choice of living space. Between-social class differences in the opportunities for choosing a living space is a potential source of spatial segregation and is reflected by a differential behavior of the pandemic.

To this point, we have assessed the extent to which the differences in social segmentation and the current urban structure have supported or weakened the expansion of the pandemic, which one the correlation of social structure, the structure of the urban space, and the evolution of the pandemic. When seeking to analyze the characteristics of COVID-19 propagation from this perspective, we cannot isolate the forms of production of urban space from the social conceptualization of space. We are convinced that the intensity of socio-spatial segregation and the forms of the urban structure have determined the intensity of the spread of the COVID-19 pandemic.

Impact data regarding the social areas shows that there are very precise relationships between the social character of the social areas and the evolution of the urban structure. We can observe differential relationships that are more characteristic in some social areas than others, such as that between very high social status and very high family status, or vice versa. On the one hand, we can identify the processes that lead to the social valuation of urban spaces, especially in relation to the uses that social groups make of the different areas of urban space. On the other hand, we can observe the relationships between the processes of expansion, growth, or reform of urban spaces and their effects on socio-spatial differentiation. Heterogeneity is a differential feature of urban spaces that has classically been used to derive or justify the features that define the urban structure and, therefore, the guidelines and values of the population groups in the social areas [70,71]. Our analysis confirms that there are large contrasts in social and family status in this urban space and there are nuanced and complex differences between social classes.

## 4. Discussion

The proposed model should be considered not just in the context of our hypothesis, but rather in a far broader sense: It can explain the functioning of relationships between the societal structure and socio-spatial segregation. More than an a priori theory, the hypothesis raised in this work reflects the aim to interpret the facts revealed by the data on infections received: that there is a direct relationship between social and spatial structures that can be synthesized through the concept of social vulnerability, and that this impacts the intensity and forms of contagion distribution in urban spaces. It is difficult to model a theory on how society and the use of space relate to the evolution of the pandemic. Such a strategy is reductionist, and thus cannot be based simply on the economic determinism of ecology. This was previously seen for the study of cultural values and in the most recent concerns about the perception or symbolic character of urban space.

The proposed model considers the four categories of family status, social status, social care, and territorial or spatial framework for factorial analysis. Each category is considered to be fundamental to the social segregation of the urban space. They can be studied separately, but better conclusions can be obtained from their joint analysis. For this reason, the interactions across the four categories will be considered in the model, because, in turn, the categories are elements of the urban spatial structure. The definition of these categories and the establishment of their interrelationships makes the model useful and will serve as the basis for analyzing the factors responsible for determining the distribution of COVID-19 infections in any urban space. Here, examining the material and social capacities of the different social areas can reveal how spatial inequalities affect the spread and management of the pandemic.

We designed this model with the hope of not only envisioning the socio-spatial structure of the urban space of Malaga, but also comparing this structure with data on the COVID-19 pandemic. By analyzing the modeled social areas, we seek to provide a basis for analyzing the differentiation of society. The theory of social areas can be used to show the essential features (categories) by which the urban space is structured into socially differentiated areas within a city (social areas). We believe that factor analysis allows us to validate the model in terms of procedure. By using this statistical method to elucidate the variables of intra-urban social differentiation, we seek to maintain the connection with the theory of the model of social areas, which was developed from the perspective of human ecology. As emphasized throughout, the analysis is oriented toward identifying certain dimensions (herein called categories) based on the theoretical model of social areas.

It would be interesting to investigate the influence of urban mobility on the transmission of infections. Our scan of the literature led us to conclude that including urban mobility in the model would have greatly increased its complexity, so we chose to exclude it and focus on the objective that we had set for ourselves. However, more studies on the subject are appearing every day [72,73,74,75].

Another aspect that seems worthy of future study is the inclusion of migratory status in the model. This could help explain the results obtained in the most vulnerable social areas identified herein. Migratory status can be used to observe how the social structure nuances that characterize a migratory population may influence the model. Even if migratory status is not directly used in the factor analysis, it can be considered as a corrective factor for those variables that allude to the vulnerability of different groups in the social areas. All this is justified in the determination of the growing scale based on the complexity of the organization of modern societies, in which the growing mobility of the population is identified. The migratory status can be used to reflect a further degree of internal heterogeneity in the social areas and allow the internal segregation of groups to be assessed based on their differentiation by origin. The possible influence of migratory status as a factor for differentiation can be verified when the heterogeneity of the population (in terms of its origin) manifests itself as a factor of spatial segregation, regardless of other motivations (e.g., economic ones). Moreover, different cultural contexts will introduce differences in the way of life of the inhabitants of urban spaces.

Internal migration is a decisive component of the spatial redistribution of a population in urban spaces and has implications for communities, households, and individuals. In the context of the pandemic, we believe that migratory movements in cities can show social effects and behaviors related to the cultural communities to which their population groups belong. For households and individuals, migration (particularly if it is part of an elaborate strategy) can have a relevant effect on the vulnerability imposed by the pandemic; migrant populations might be among the most affected groups and therefore in need of care, along with other vulnerable groups of the local population.

This work raises several interesting thoughts. Some relate to the configuration of urban space as a determinant of the living conditions of populations, and its manifestation in the social contrast-related differences in the intensities of COVID-19 infection and spreading. The higher impact of the pandemic on areas of low social status and advanced life cycle emphasizes the importance of the social differences that can be observed within urban spaces. This suggests that the characteristics of urban society (the sum of all its areas) should be critically considered when seeking to establish control measures for a pandemic.

Finally, we think that the knowledge of the socio-spatial aspects that characterize the social areas can provide very useful information for the establishment of information policies and prevention strategies in health (health literacy) [76,77,78,79,80,81,82,83], which would be convenient to be raised or directed, where appropriate, attending to this characterization so that the most optimal effect possible is achieved. The same policy will not have similar results throughout the urban space and will produce different levels of acceptance and compliance depending on the social characteristics of the groups that inhabit them. Different authors have worked on this issue and concluded that health literacy is related to the level of literacy and training of people, which implies skills and competences to access and understand information about illness, prevention, and health promotion, in order to apply it in their daily lives and thereby maintain or improve their quality of life, and, in the specific case that we are faced with, in the prevention or monitoring of mitigation measures in order to minimize the transmission of a pandemic, such as that generated by COVID-19 [84,85,86].

People’s need for information has increased significantly in the time of COVID-19. They face challenges and barriers when trying to find relevant training for them, to critically reflect upon information, and use such information to make health decisions in their daily lives. In this context, people with higher levels of education will more easily identify individual and social problems and will handle health-related issues with greater skill and opportunity [87,88,89]. On the contrary, several authors point out that a deficient health literacy of the population will imply a lower capacity to apply the precepts that should govern a healthy life, that the levels of perceived health will be lower, that the precepts established by the preventive services will not be addressed with sufficient acceptance, which will lead to an excessive frequentation of health services (for example: use emergency hospital services as primary care services), worse knowledge of treatments in cases of chronic diseases, and, in short, a greater possibility of hospitalization [90,91].

## 5. Conclusions

In this study, we start by constructing synthetic indices of different dimensions of vulnerability for an urban space, as seen prior to the outbreak of the COVID-19 pandemic. Using these indices along with complex homogenization methods, we synthesize socioeconomic information from different origins and geographical references into four categories. We show that urban spaces of Malaga, Spain, before the pandemic exhibited disparate degrees of social, family, care (most evident), and/or territorial vulnerability. We also identify the most vulnerable spaces according to each category.

Considering the relationship between the degree of vulnerability and the spread of COVID-19, we identify a double expansion pattern. Initially, the most highly COVID-19-affected areas did not correspond to the most vulnerable districts or social areas. In fact, the opposite is seen. As control mechanisms began to be established based on increasing knowledge of the dynamics of COVID-19, however, the density of infections by social areas increasingly showed significant associations with the degree of vulnerability. Eventually, it became clear that vulnerability played an important role in the infection density of each social area.

The general discourse holds that there have been successive waves of the pandemic. Here, we further show that the extraordinary infective capacity of COVID-19 can vary according to the defining characteristics of each social space of an urban environment. Indeed, the mitigation measures that have been carried out by health authorities (and have generated significant social problems or rejection) have yielded differential results due to the differences, complexity, and social heterogeneity found in the different urban spaces.

Our results show that the pandemic (at least up to the point studied herein) is related to social vulnerability. Going forward, we need to continue studying the expansion of the pandemic and seeking to correlate the density of infections with the vulnerability of the urban spaces. In addition, the observed evolution of the association between the density of infections and social vulnerability could help to inform preventive management decisions.

The analysis presented herein should not be viewed as being limited to the pandemic. It will be interesting to analyze the social vulnerability of urban spaces “post-COVID-19”. This analysis would allow us to assess how the pandemic impacted the geographic distribution and intensity of the vulnerability indicators in different districts within the same urban space, as well as the resilience capacity of each.

The information derived from the analysis of the geographical distribution of the pandemic will undoubtedly be needed for the implementation of inclusive recovery policies that aim to care for the most vulnerable areas in the future.

## Figures and Tables

**Figure 1 ijerph-19-05336-f001:**
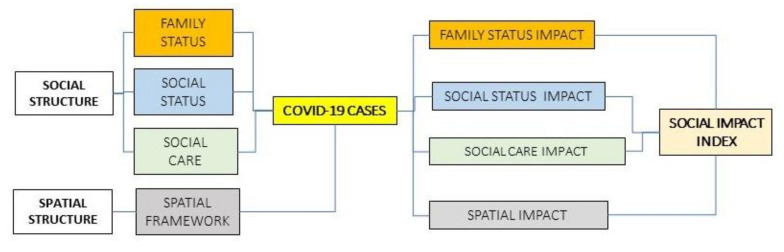
Scheme of the proposed model. Source: Author.

**Figure 2 ijerph-19-05336-f002:**
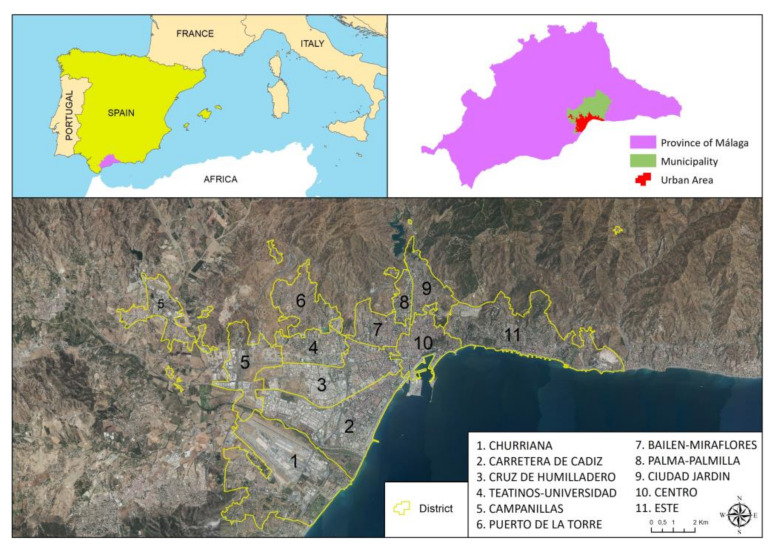
Location map of the study area and definition of urban areas. The numbered, yellow-outlined areas indicate districts, which are characterized by homogeneous social behaviors. Source: Author.

**Figure 3 ijerph-19-05336-f003:**
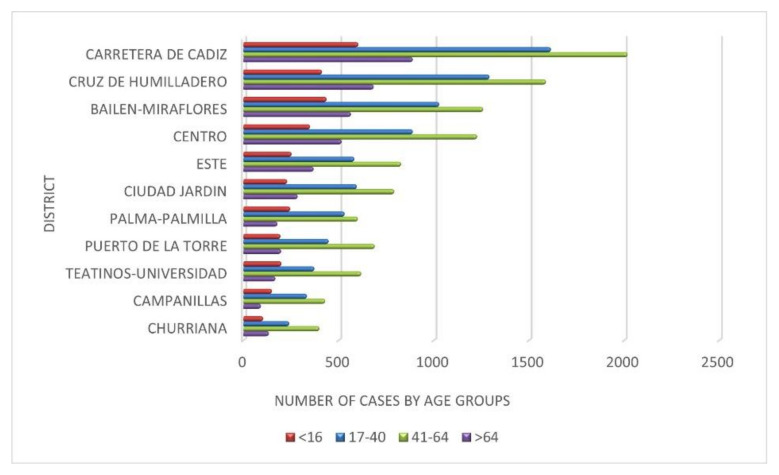
Number of infections by large age groups: young people “<16 years”, young adults “17–40”, adults “41–64” and the elderly “>64)”in social areas. Source: Author’s elaboration based on data from the Technical Subdirectorate of Information Management of the Andalusian Health Service (SAS), Regional Government.

**Figure 4 ijerph-19-05336-f004:**
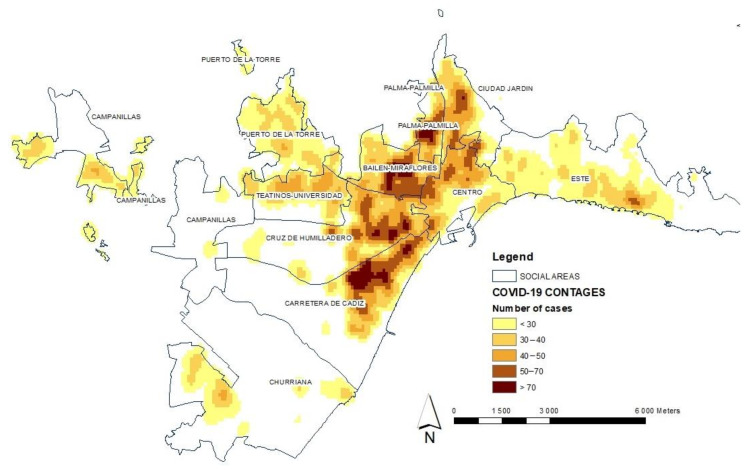
Analysis of the distribution of total infections by COVID-19 in the social areas of the city of Malaga during the period from 5 March 2020 to 31 March 2021. The behavior of the pandemic is observed through the cases of contagion in the districts of the city. Source: Author’s elaboration based on data from the Technical Subdirectorate of Information Management of the Andalusian Health Service (SAS), Regional Government.

**Figure 5 ijerph-19-05336-f005:**
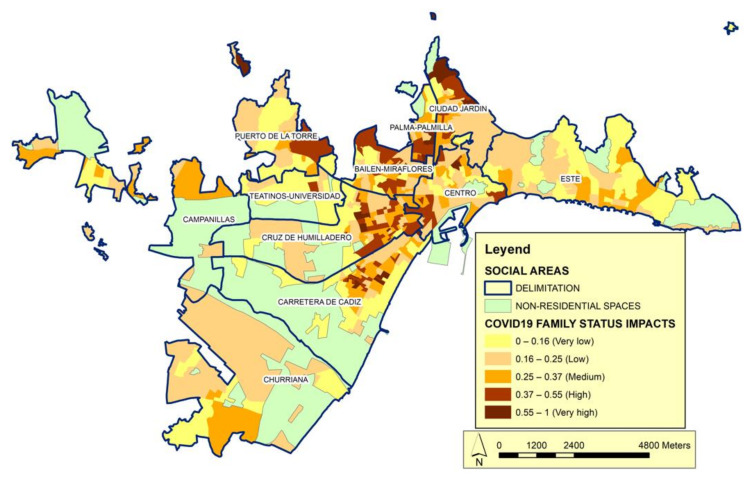
Distribution of the impact of COVID-19 according to the Family Status category. Source: Author’s elaboration. The impact index shows the correlation between Family Status vulnerability calculated according to the variables of this category in relation to the number of cases.

**Figure 6 ijerph-19-05336-f006:**
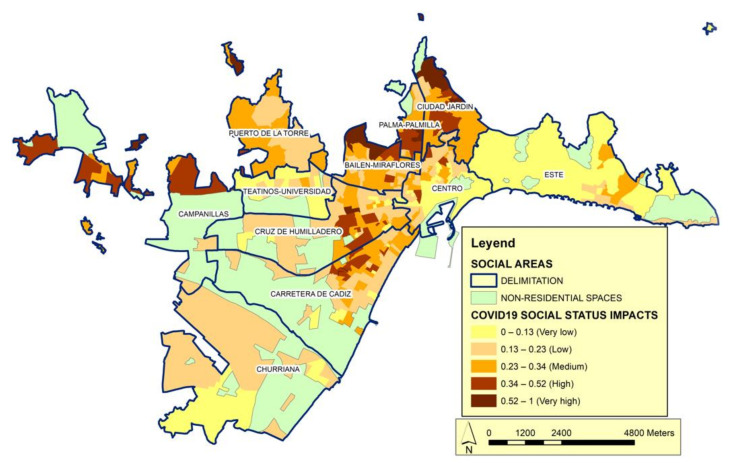
Distribution of the impact of COVID-19 according to the Social Status category. Source: Author’s elaboration. The impact index shows the correlation between Social Status vulnerability calculated according to the variables of this category in relation to the number of cases.

**Figure 7 ijerph-19-05336-f007:**
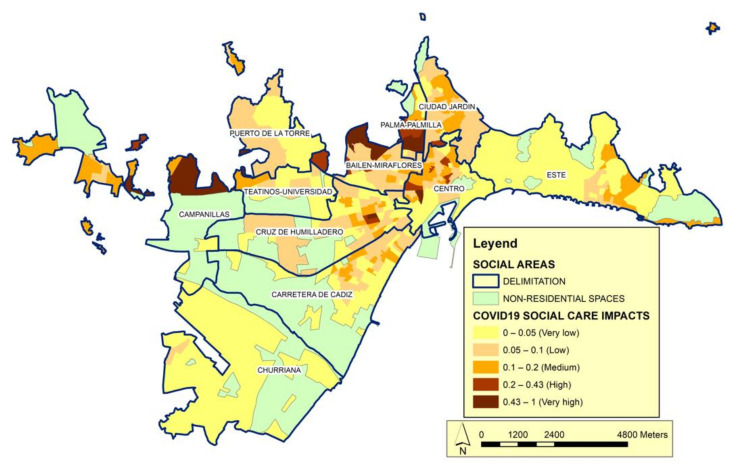
Distribution of the impact of COVID-19 according to the Social Care category. Source: Author’s elaboration. The impact index shows the correlation between Social Care vulnerability calculated according to the variables of this category in relation to the number of cases.

**Figure 8 ijerph-19-05336-f008:**
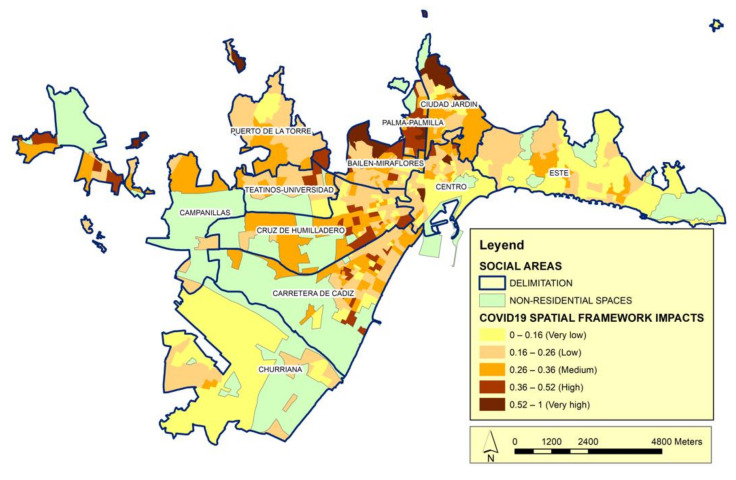
Distribution of the impact of COVID-19 according to the Spatial Framework category. Source: Author’s elaboration. The impact index shows the correlation between Spatial Framework vulnerability calculated according to the variables of this category in relation to the number of cases.

**Figure 9 ijerph-19-05336-f009:**
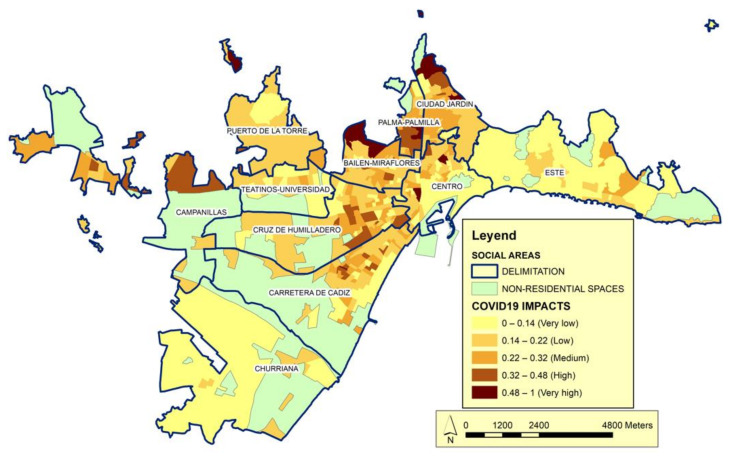
Global Impact Index. Source: Author’s elaboration. The Global Impact Index shows the correlation between the impact indices of the four categories: family status, social status, social care and spatial framework.

**Figure 10 ijerph-19-05336-f010:**
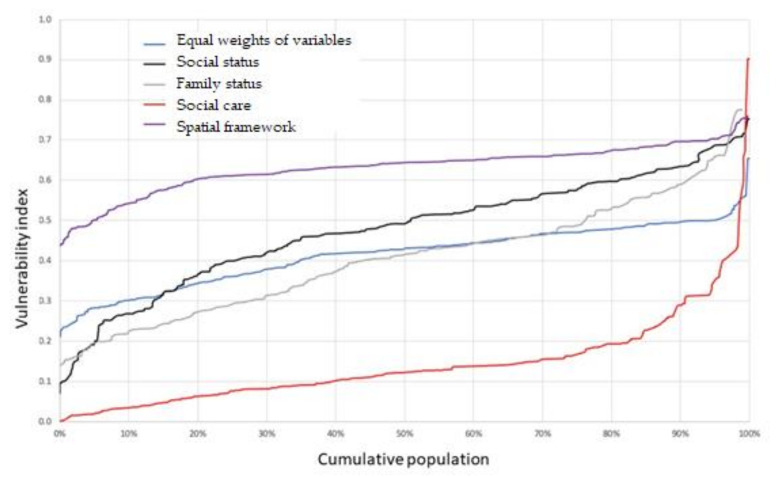
Pen’s Parade curve of the vulnerability index for the population, by category. Source: Author.

**Table 1 ijerph-19-05336-t001:** Summary table of the characteristics of the urban structure (Ordered by descending number of COVID-19 cases). Source: Municipal Register of Inhabitants (National Institute of Statistics [INE] Government of Spain), year 2020, and Information Management of the Andalusian Health Service (SAS) Regional Government (Spain). Cases from 5 March 2020 to 31 March 2021. We have preferred to keep the locally known name for the district for better identification.

Urban Structure	District (Social Area)	Population	Area km^2^	Density Pop./km^2^	Cases COVID-19	% Cases COVID-19
Historic Urban (Post 1960)	Carretera de Cádiz	115,686	13.57	8527.97	5113	4.42%
Historic Urban (Post 1950)	Cruz de Humilladero	88,996	10.28	8655.20	3965	4.46%
Historic Urban (Post 1960)	Bailen-Miraflores	68,965	3.89	17,728.46	3275	4.75%
Historic Center (Post 1850)	Centro Histórico	80,254	5.08	15,809.89	2968	3.70%
Historic Urban (Post 1970)	Málaga Este	55,473	14.28	3884.69	2017	3.64%
Historic Urban (Post 1940)	Ciudad Jardín	38,602	4.44	8693.77	1888	4.89%
Historic Urban (Post 1970)	Palma-Palmilla	26,286	1.87	14,076.20	1542	5.87%
Urban Periphery (Post 1980)	Puerto de la Torre	34,883	5.63	6196.34	1515	4.34%
Recent Urban (Post 1990)	Teatinos-Universidad	26,711	4.42	6039.69	1344	5.03%
Recent Urban (Post 1980)	Campanillas	18,008	11.7	1538.63	990	5.50%
Urban Periphery (Post 1980)	Churriana	20,387	19.3	1056.55	862	4.23%
	STUDY AREA	574,251	94.46	8382.49	25,479	4.44%

**Table 2 ijerph-19-05336-t002:** Summary of the categories of the model and the data sources for each. Source: Authors.

Category	Source	Date
Family Status	Municipal Register of Inhabitants, National Institute of Statistics, Government of Spain	2019
Social Status	Survey of Living Conditions (ECV) of the National Institute of Statistics, Government of Spain	2019
Social Care	Information System for Users of Social Services (SIUSS-Junta de Andalucía) Regional Government	2019
Spatial Framework	Malaga Urban Environment Observatory, Malaga City Council	2020

**Table 3 ijerph-19-05336-t003:** COVID contagions and vulnerability indices, global and for the four categories, for every Districts (Sorted ascending by the Global Vulnerability Index). Source: Authors.

District	Cases COVID-19	Global Vulnerability Index	Subindex: Familiar Status	Subindex: Social Status	Subindex: Social Care	Subindex: Spatial Framework
Málaga Este	2017	0.2939	0.1940	0.2736	0.0445	0.6635
Palma-Palmilla	1542	0.3061	0.2808	0.3314	0.0417	0.5706
Carretera de Cádiz	5113	0.3396	0.3568	0.2954	0.0801	0.6262
Centro Histórico	2968	0.3695	0.2680	0.4584	0.1137	0.6378
Bailén-Miraflores	3275	0.4209	0.3823	0.5355	0.0964	0.6694
Churriana	862	0.4244	0.4150	0.4410	0.1698	0.6720
Teatinos-Universidad	1344	0.4259	0.3831	0.5429	0.1175	0.6602
Puerto de la Torre	1515	0.4324	0.4164	0.5221	0.1199	0.6711
Ciudad Jardín	1888	0.4477	0.2619	0.6154	0.2113	0.7020
Cruz de Humilladero	3965	0.4485	0.3721	0.6377	0.1446	0.6396
Campanillas	990	0.4629	0.3096	0.5681	0.2624	0.7116

**Table 4 ijerph-19-05336-t004:** Total number of infections, hospital admissions, and deaths by social area. The total population is included for reference. Source: Author’s elaboration based on data from the Technical Subdirectorate of Information Management of the Andalusian Health Service (SAS), Regional Government.

Urban Structure	District	Population	Number of Infections	%	Hospital Admissions	%	Number of Deceased	%	Average Age
Historic urban (post 1960)	Carretera de Cádiz	115,686	5043	19.79	478	19.53	77	15.34	46.06
Historic urban (post 1940)	Cruz de Humilladero	88,996	4102	16.10	463	18.92	86	17.13	46.58
Historic urban (post 1960)	Bailen-Miraflo- res	68,965	3258	12.79	330	13.49	93	18.53	46.66
Historic center (post 1850)	Centro	80,254	2918	11.45	271	11.07	78	15.54	47.87
Historic urban (post 1970	Málaga Este	55,473	2037	7.99	191	7.81	49	9.76	48.05
Historic urban (post 1940)	Ciudad Jardín	38,602	1943	7.63	153	6.25	38	7.57	46.55
Urban periphery (post 1980)	Puerto de la Torre	34,883	1562	6.13	169	6.91	21	4.18	46.61
Recent urban (post 1990)	Teatinos-Universi- dad	26,711	1314	5.16	130	5.31	17	3.39	43.27
Historic urban (post 1970)	Palma-Palmilla	26,286	1510	5.93	114	4.66	19	3.78	43.52
Urban periphery (post 1980)	Churriana	20,387	890	3.49	81	3.31	7	1.39	48.09
Recent urban (post 1980)	Campanillas	18,008	902	3.54	67	2.74	17	3.39	42.15
	TOTAL	574,251	25,479	100	2447	100	502	100	45.95

## Data Availability

Not applicable.

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
