# Peer review of "Evidence of the Relationship between Social Vulnerability and the Spread of COVID-19 in Urban Spaces"

_ijerph, 2022, doi:10.3390/ijerph19095336_

Round 1
Reviewer 1 Report
This article correctly expands the theories on social vulnerability. The research proposes a very interesting model that can be replicated in other places and in the face of other phenomena. It does a good job of synthesis and a good spatial analysis of the city of Malaga.
The authors hypothesize, by visualizing the unequal incidence of COVID-19 disease in different urban spaces, that an unequal effect of the pandemic occurred to the different areas. The hypothesis is correctly corroborated at the end. To this end, the selection of the four categories for the factor analysis (family status, social status, social care and territorial or spatial framework) is very pertinent.
The only problem I found was related to the creation of the indexes. Although it might have been an oversight on my part, I haven’t been able to discern how they calculated the results in Table 3. It is important to clarify this aspect in order to be able to replicate the study in other places. I don’t think that this clarification takes away from their main argument nor their results.
The authors do not need to integrate these changes if they do not wish to.
This study has great applied value.
Overall, I thought this was an informative paper that highlights a lesser-known aspect of COVID studies. I enjoyed reading the insights the authors gleaned from their research.
Author Response
Thank you very much for your fruitful comments, that drove us to a better version. In particular, we included some more insight related with this comment:
- “The only problem I found was related to the creation of the indexes. Although it might have been an oversight on my part, I haven’t been able to discern how they calculated the results in Table 3. It is important to clarify this aspect in order to be able to replicate the study in other places. I don’t think that this clarification takes away from their main argument nor their results.”
We included some more explanations on the indexes’ construction and details on page 5. Lines 172-177.
Please see the attachment.

Reviewer 2 Report
The manuscript speaks about the relationship between social vulnerability and the spread of Covid-19 pandemic. A Spanish case study was carried out.
In the hard times we are living, this is an interesting topic which can be also generalised speaking about social inequalities and healthcare systems, as well as I consider this manuscript a good opportunity to assess the phenomenon of health literacy, which is damaged by social inequalities as well. I would recommend the autor(s) to follow up in the discussion remarks the phenomenon. Here some references which may help:
D. Nutbeam Health literacy as a public health goal: a challenge for contemporary health education and communication strategies into the 21st century Health Promot. Int., 15 (3) (2000), pp. 259-267, 10.1093/heapro/15.3.259 2021 "The more you know, the better you act? Institutional communication in Covid-19 crisis management" Technological Forecasting and Social Change 170 May 120929 DOI: https://doi.org/10.1016/j.techfore.2021.120929 As it concerns the style of the manuscript, I would recommend to split introduction and create a section of theoretical background. No issues regarding the quality of language have been detected. I would suggest a major revision. But the work is worthy of publication.Author Response
Thank you very much for your fruitful comments. We have included the suggested references, that we found so interesting, and suitable for our work.
A reference to this effect has been introduced in the text, in the discussion sections. Lines 581-606.
Please see the attachment.

Reviewer 3 Report
Intriguing general concept for the paper. Some points for clarification, though.
Table 1: Please include % for cases of COVID-19 against population.
p.4: 'family status, social status, social care, and spatial framework vulnerability' subindices are not well defined.
p. 6: Why only one range for adults? This makes the figure suspect because it might be assumed that groups are equal size. It's hard to tell what if anything we should take from the figure.
Speaking of which, is there a chance that certain ages (like 42-47 - which is a range not an 'average age') behaved differently and this explains incidence?
p. 7: figure 4 - does this correspond to population generally?
"more cases of contagion in some areas than in other" - but is this expected?
p. 8: In order to understand this - the indices need to be made clearer.
"Social status is largely deter
mined by the characteristic of household income, through which the average annual net 258
income is quantified by type of household to provide information on the distribution of 259
income. " - why not use income then?
Elaboration on the indices and how they are constructed, and what they mean, would be helpful.
p. 11: "These areas are not socially mar
ginal, but rather present below-average social and family traits." I'm not sure what this means.
Same page: "the social status is medium-high" - fine for differentiation purposes but what are we to take from that? What practical difference does it make?
The paper is potentially useful but the analysis/interpretation is not clear. It seems like a lot is being assumed from the way this is presented.
Author Response
Thank you very much for your fruitful comments, that were so useful to improve the paper. Below, we detail how we modified the paper according to each of them:
- “Table 1: Please include % for cases of COVID-19 against population.”
Done.
- “4: 'family status, social status, social care, and spatial framework vulnerability' subindices are not well defined.”
We have included, in page 5, more details on the construction of the indices, as well as their interpretation.
- “p. 6: Why only one range for adults? This makes the figure suspect because it might be assumed that groups are equal size. It's hard to tell what if anything we should take from the figure. Speaking of which, is there a chance that certain ages (like 42-47 - which is a range not an 'average age') behaved differently and this explains incidence?”
The inclusion of age ranges has been due to the fact that they are the age ranges used by statistical agencies in Spain. However, as this may involve some kind of confusion we have divided the age range of adults into two groups, to differentiate the youngest adults (17 to 40 years) from the more mature adults in age (41 to 64 years).
This paragraph It is redrafted. Lines 201-224.
- “p. 7: figure 4 - does this correspond to population generally? "more cases of contagion in some areas than in other" - but is this expected?“
This map has been made taking into account the location of the total number of cases of infections without distinction of ages and in the period analyzed considering each case as a specific element. To this end, the information of each affected person has been obtained without distinction of age or other conditions, only considering their affectation and their geolocation.
Please see the explanation in lines 237-251.
- “p. 8: In order to understand this - the indices need to be made clearer. "Social status is largely deter mined by the characteristic of household income, through which the average annual net income is quantified by type of household to provide information on the distribution of " - why not use income then? Elaboration on the indices and how they are constructed, and what they mean, would be helpful.”
This paragraph has been redrafted: lines 289-306.
- 11: "These areas are not socially mar ginal, but rather present below-average social and family traits." I'm not sure what this means. Same page: "the social status is medium-high" - fine for differentiation purposes but what are we to take from that? What practical difference does it make? The paper is potentially useful but the analysis/interpretation is not clear. It seems like a lot is being assumed from the way this is presented.
The statement “These areas are not socially marginal, but rather present below-average social and family traits” was modified to be clearer: “These areas, without being on the extreme vulnerability, are on the highest vulnerability segments.” Similarly, the statement “the social status is medium-high” was modified to: “the vulnerability level is medium-low”
Added a new explanation. Lines 423-444.
Please see the attachment.

Round 2
Reviewer 2 Report
I appreciate the profound work of revision conducted by the author(s), so they provided new useful insights and elements of analysis which make the paper eligible for full acceptance for publication.
Reviewer 3 Report
My concerns have been sufficiently addressed by the responses of the author team.